# Migration of Reflector Orientation Attributes in Deep Seismic Profiles: Evidence for Decoupling of the Yilgarn Craton Lower Crust

Andrew J. Calvert[1] and Michael P. Doublier[2,3]

[1]Department of Earth Sciences, Simon Fraser University, Burnaby, British Columbia, V5A 1S6, Canada
[2]Mineral Systems Branch, Geoscience Australia, Symonston, ACT 2609, Australia
[3]Centre for Exploration Targeting, School of Earth and Environment, University of Western Australia, Crawley, WA 6009, Australia

*Correspondence to*: Andrew J. Calvert (acalvert@sfu.ca)

**Abstract.** Interpretation of deep seismic data is challenging due to the lack of direct geological constraints from drilling and the more limited amount of data available from 2-D profiles in comparison to hydrocarbon exploration surveys. Thus other constraints that can be derived from the seismic data themselves can be of great value. Though the origin of most deep seismic reflections remains ambiguous, an association between seismic reflections and crustal strain, e.g. shear zones, underlies many interpretations. Estimates of the 3D orientation of reflectors may help associate specific reflections, or regions of the crust, with geological structures mapped at the surface whose orientation and tectonic history are known. In the case of crooked 2-D onshore seismic lines, the orientation of reflections can be estimated when the range of azimuths in a common midpoint gather is greater than approximately 20 degrees, but integration of these local orientation attributes into an interpretation of migrated seismic data requires that they also be migrated. Here we present a simple approach to the 2-D migration of these orientation attributes that utilises the apparent dip of reflections on the unmigrated stack, and maps reflector strike, for example, to a short linear segment depending on its original position and a migration velocity. This interpretation approach has been applied to a seismic line shot across the Younami Terrane of the Australian Yilgarn Craton, and indicates that the lower crust behaved differently from the overlying middle crust as the newly assembled crust collapsed during the Late Archean. Some structures related to approximately east-directed shortening are preserved in the middle crust, but the lower crust is characterized by reflectors that suggest N-NNE-oriented ductile flow. Deployment of off-line receivers during seismic acquisition allows the recording of a larger range of source-receiver azimuths, and should produce more reliable future estimates of these reflector attributes.

## 1 Introduction

Deep seismic reflection surveys that image the entire continental crust are typically acquired as 2-D profiles due to cost, and are able to provide sub-surface images with a resolution of the order of 100 m or better. The interpretation of these deep seismic profiles, however, is often limited by the presence of reflections that can originate from locations out of the plane of the seismic profile, resulting in cross-cutting reflections in the migrated seismic section. In such situations it is difficult to identify which

reflection, if any, should be included in an interpretation. Many onshore profiles have a crooked geometry because they are acquired along existing access roads. By using a 3-D travel time equation to determine the coherence of reflections, Bellefleur et al. (1997) showed how this limited 3-D geometry could be exploited to estimate the true 3-D orientation of subsurface reflectors where the acquisition line was particularly crooked, for example at sudden large bends in the road. Taking advantage

of the increase in computing power over the last two decades, Calvert (2017) extended this method to every common depth point (CDP) in a crooked seismic profile, additionally providing quantitative estimates of the relative errors in the estimated angles of reflector dip and strike, and potentially also stacking velocity. These results, for example the angles and error estimates, are displayed as a function of time at each CDP on unmigrated seismic sections. Although it is possible to make general inferences on the distribution of subsurface reflectors, more detailed interpretation requires that the angle estimates be

represented closer to their true subsurface position, i.e. on migrated seismic sections, which is an issue that was not addressed by Calvert (2017). The purpose of this paper is to present an approach to the migration of these reflector orientation attributes that allows their use in the interpretation of conventional 2-D migrated deep seismic sections; for example, by migrating more steeply dipping reflections into the middle crust, the predominant orientation of lower crustal reflections can be clarified. The importance of obtaining more accurate orientation estimates for positioning reflectors in 3-D, by for example deploying

additional cross-line receivers, will also be discussed.

## 2 Reflector Orientation Estimation

When a crooked seismic reflection line is processed, it is necessary to choose a slalom line through the distribution of source-receiver midpoints, and to define the CDP bins together with their dimensions along this line. Within a CDP bin, the conventional 2-D hyperbolic travel time equation may not accurately represent out-of-plane reflections due to the varying

source-receiver azimuths. In these circumstances and under the straight-ray assumption used in stacking velocity analysis (Taner and Koehler, 1969), reflection travel times are better described by a 3-D travel time equation that includes the dip and strike of the reflector (Levin, 1971). When the seismic line is linear, the angles representing dip and strike cannot be uniquely determined, but along a crooked seismic profile, the distribution of source-receiver azimuths within a CDP gather varies, allowing the dip and strike to be well determined if a sufficiently large range of azimuths is present, for example where there

is a large change in the direction of a seismic line (Bellefleur et al., 1997). In practice, most single CDP gathers on a crooked seismic line contain an insufficient number of traces, but this limitation can be mostly overcome by combining multiple CDP gathers into a much larger supergather that can be used for the estimation process; both Bellefleur et al. (1997) and Calvert (2017) provide examples of how the use of a large supergather permits the independent recovery of both dip and strike angles in many situations. The estimation method assumes that reflections within the supergather originate from a locally planar

interface; as more CDP gathers are combined, this assumption can break down, especially where the geology is complex, for example where folded reflectors are present; for the crooked lines tested, supergathers of 40-80 CDPs appear to be adequate. If the algorithm were applied to every CDP gather with the output comprising the stacked trace computed using a moveout

correction based on the estimated values of dip and strike, then this process could be viewed as an automated version of the crossdip correction that is often applied manually to crooked seismic profiles , e.g. Nedimovoc and West (2003a) or Beckel and Juhlin (2018). It should, however, be noted that this crossdip correction usually makes the assumption of linear moveout in the cross-dip direction within a CDP gather, which is not necessarily the case, especially where the line is particularly

crooked.

Thus in a CDP gather and assuming a root mean square (RMS) velocity function, the semblance of a reflection (Neidell and Taner, 1969) can be calculated using a small time window, e.g. 40 ms, at each zero offset time for a range of trial angles of dip and strike, which is measured from north. At each zero offset time, the estimated dip and strike correspond to the angles

with the maximum semblance, i.e. the most coherent reflection (Bellefleur et al., 1997). Although the searched strike angle varies from $-180^o$ to $+180^o$, only values between $0^o$ and $180^o$ are output in the algorithm employed here because negative values are increased by $180^o$ to ensure that the same value is output for reflectors with parallel strike directions but an opposite sense of dip; for example, reflectors dipping to the north and south will both be assigned a strike of $90^o$ (Calvert, 2017). Since this method is a grid search, the relative error in the orientation angles can be characterized by defining a threshold, for example

90% of the maximum semblance, and finding the largest difference in angle from this maximum to any other angle with a semblance greater than the threshold. These error values characterize at each time sample the size of the semblance maximum as a function of dip and strike; as an example, for a horizontal reflector and a survey geometry with a broad range of source-receiver azimuths, the dip angle will likely be well resolved, but the error for the estimated strike could be as large as $\pm 90^o$, because the strike is not well-defined in this specific case. It is additionally possible to extend the method to velocity analysis

by repeating the estimation of an optimal dip and strike angle for a range of trial RMS velocity values; more details of the error estimation can be found in Calvert (2017). Since all the estimated attributes, angle, velocity, and error, are found for each zero offset time sample within a CDP gather, they are represented on a seismic section that corresponds to the unmigrated stack, i.e. the attributes are not positioned at their true subsurface position.

## 2.1 Attribute Migration

It is possible to apply 3-D prestack time migration to crooked 2-D reflection profiles; in some cases, for example where the deviation from 2-D is not great, the result is readily interpretable, but in others, the output 3-D volume can be dominated by artefacts from wave equation migration with most structures incompletely imaged due to the limited amount of data recorded in the cross-line direction (Nedimovic and West, 2003b). Thus crooked 2-D seismic profiles are usually migrated in 2-D for interpretation. To better integrate orientation angle estimates into the seismic interpretation it is therefore desirable to reposition

these attributes in a way that is analagous to seismic data migration, so they can be superimposed on their corresponding reflections. Attributes do not satisfy the assumptions necessary for wave-equation migration, and the result of applying such an algorithm to an unmigrated section containing attribute values would be meaningless. However, if the apparent dip of reflections on the unmigrated section is known, then a line migration or segment migration algorithm can be used to position

the attribute value at a new output location corresponding to the migrated position of the corresponding reflection (Hagedoorn, 1954; Calvert, 2004). The sample value at each time and CDP location on the unmigrated section can be mapped to a small linear segment whose output position and dip is determined by the input position, apparent dip, and migration velocity (Reynaud, 1988). With seismic data, when multiple reflections are mapped to the same output location they are summed together, but for the attribute migration algorithm presented here, the output value is modified to be the attribute with the greatest semblance, implying that some less coherent attribute values will not be represented in the migrated output.

In principle, input attributes could be mapped to planar facets within a 3-D volume, but with narrow-azimuth crooked line surveys the uncertainties in determining the dip and strike along individual reflections are likely to be too large, resulting in the fragmentation of individual reflections after migration. Though 3-D migration is preferable in theory, the interpretation of an incomplete, sparse set of reflections in a 3-D volume is also likely to be challenging, and a better approach may be forward modelling the appearance of 3-D structures in the crooked 2-D seismic profile.

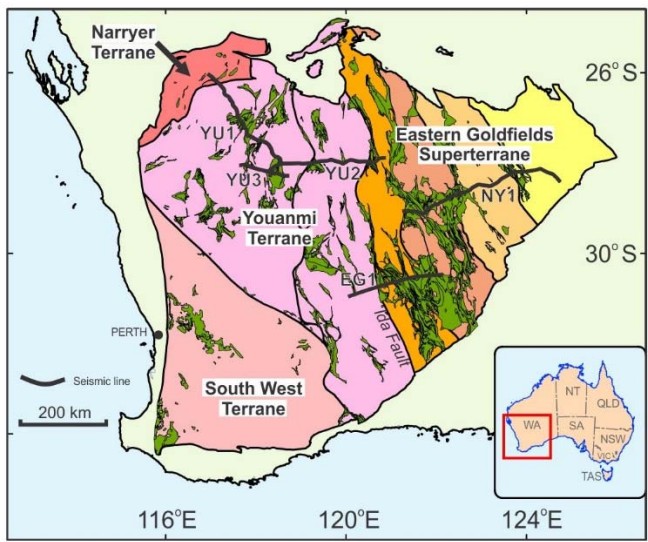

**Figure 1: Major terranes of the Yilgarn Craton in Australia with locations of deep seismic lines. The Youanmi Terrane represents the older core of the craton to which terranes of the Eastern Goldfields Superterrane were accreted during the late Archean. Greenstone belts are shown in green.**

## 3 Yilgarn Craton Example

This pragmatic approach to attribute migration is illustrated using a high quality seismic line, 10GA-YU2, which was shot in 2010 over the Youanmi Terrane of the Archean Yilgarn Craton of Western Australia as a collaborative project between Geoscience Australia and the Geological Survey of Western Australia (Fig. 1; Wyche et al. 2013). The Youanmi Terrane, which contains several north-northeast striking greenstone belts, is the 3.05-2.70 Ga core of the craton (Pigeon and Wilde,

1990; Van Kranendonk et al., 2013). It is separated by the Ida Fault from the >2.95-2.66 Ga Eastern Goldfields Superterrane (Czarnota et al., 2010), which was accreted during a period of intermittent crustal shortening from >2.73 Ga to 2.65 Ga (Myers, 1995). The seismic line extends from the 2.81 Ga Windimurra Igneous Complex (Ivanic et al. 2013) in the west into the Eastern Goldfields Superterrane in the east, and is mostly located over granitoid plutons and tonalitic gneiss, but also crosses the Sandstone greenstone belt. The interior of the Yilgarn Craton was unaffected by any large-scale post-Archean tectonic events, but was intruded by four sets of mafic dyke swarms during the Proterozoic.

## 3.1 Youanmi Seismic Survey

Line 10GA-YU2 was shot every 80 m using a source array of three Hemi60 vibrators and recorded by a 300-channel symmetric split spread with receiver groups every 40 m. A 12 s long varisweep was used with either two or three sweeps recorded at each vibration point (VP). The seismic data were originally processed by Geoscience Australia (GA) using a conventional sequence of crooked-line geometry, refraction statics, geometric spreading correction, spectral equalisation, velocity analysis, normal moveout, residuals statics, dip moveout correction, stretch mute, stack, and Kirchhoff migration; further details on the seismic acquisition and processing is provided by Costelloe and Jones (2013).

## 3.2 Reflector Orientation Estimation and Migration

The preprocessing of the prestack seismic data for orientation analysis included resampling to 8 ms, refraction statics, residual statics, amplitude recovery with a $T^{1.2}$ gain (to 12 s), time-variant spectral whitening, automatic gain control (AGC) with a 0.5 s window, zero phase Ormsby filtering to 5-10-30-40 Hz, trace muting, and combination of 64 adjacent CDP into supergathers every 2 CDP; an additional mute of data stretched more than 30% is included in the orientation estimation analysis. The reflector orientation analysis was performed on each supergather using a 56 ms time window every three degrees of dip and three degrees of strike, using a RMS velocity function that increased from 6000 m/s at 0.0 s to 6500 m/s at 12.0 s, and to 7250 m/s at 20.0 s. At each time sample, an estimate of the dip and strike of the most coherent reflection in the prestack data is obtained, together with an estimate of the relative error (Calvert, 2017).

Values of local reflector strike can complement an interpretation based on a conventional seismic section, and estimates of reflector strike along line 10GA-YU2 are shown in Figure 2a, but only for reflections with a semblance greater than 0.005 and for which the error in estimated strike angle is less than 30º, in order to remove less reliable estimates. Where the seismic line is almost linear, reflector orientation cannot be estimated accurately, and these large errors, which are shown in Figure 2b, result in the vertical white, no-data bands in Figure 2a. The error depends on the distribution of sources and receivers in the supergather used for the estimate, and their relation to the CDP bin centre; however, in practice, those parts of the seismic line where it is difficult to obtain orientation angles are reasonably well predicted by the range of useful source-receiver azimuths, which is defined to be the number of one degree azimuth bins for which there are seismic data

available (Fig. 2). This definition was chosen to account for a (perhaps unlikely) situation in which a single orthogonal trace could result in a large range of source-receiver azimuths, but would not contribute significantly to the reflector orientation estimate due to a low signal-to-noise ratio. With the geometry of this seismic line, an azimuth range greater than ~20° seems sufficient to obtain most strike estimates, but 30° is a more preferable minimum. Though most strike estimates with large

uncertainties have been excluded, there remain some parts of the seismic line where strike values are judged to be unreliable. The most evident areas are where very similar strike values extend through much of the crust in a vertical column on the unmigrated section; for example, the ~60° values (green in Fig. 2a) between 5 s and 8.5 s at CDP 14000 visible and ~50° values (yellow-green in Fig. 2a) from 6-12 s at CDP 6150.

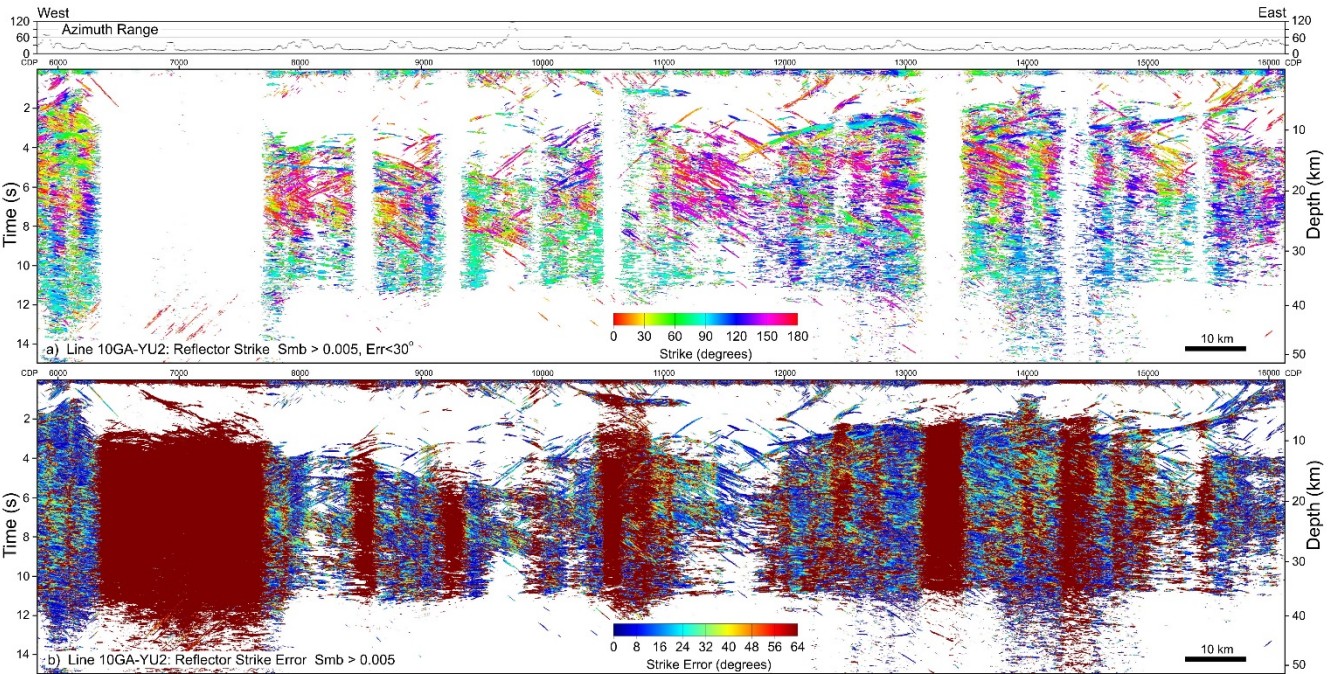

**Figure 2: (a) Unmigrated reflector strike estimated for line 10GA-YU2. The range of source-receiver azimuths available for the orientation analysis is indicated in degrees in the overlying panel. Vertical white bands correspond to unreliable values that have been excluded due to low reflection semblance or high angular uncertainty. (b) Relative error in estimated strike, i.e. the range of angles within 90% of the global semblance maximum.**

2-D migration of an orientation attribute requires two input datasets: the attribute and an estimate of the apparent dip. To ensure consistency with the conventional migration, the apparent dip was estimated from the GA-processed stack section by determining the most coherent dip in a local slant stack across an 800 m window at each time sample and CDP (Calvert, 2004). Using these apparent dips and the 1-D stacking velocity function, each attribute sample was migrated to a 320 m-long linear segment centred on its output location with only the most coherent event retained at each position, as described earlier.

The length of the output segment was selected to provide a degree of overlap for points migrated from the same reflection, creating some continuity on the output image without producing long linear segments that would be incapable of mimicking

the geometry of a curved reflector. The trace spacing of the input datasets was 40 m, and dips greater than 50° were excluded to remove some steeply dipping coherent noise present in the data. Migrated reflector strike is shown in Figure 3b, with the repositioning of reflector strike due to the migration process clear from a comparison with Figure 2a; moderately dipping events have moved updip to earlier times where they exhibit a shorter length; reflections with a strike of 0° (coloured red)

5    that occur at times of 9-11 s in the lower crust on the unmigrated section have moved into the middle crust after migration; other reflections have moved into the vertical white bands where strike values could not be well determined. After migration, anomalous columns of similar strike are much harder to identify due to the differential movement of strike values associated with different apparent dips.

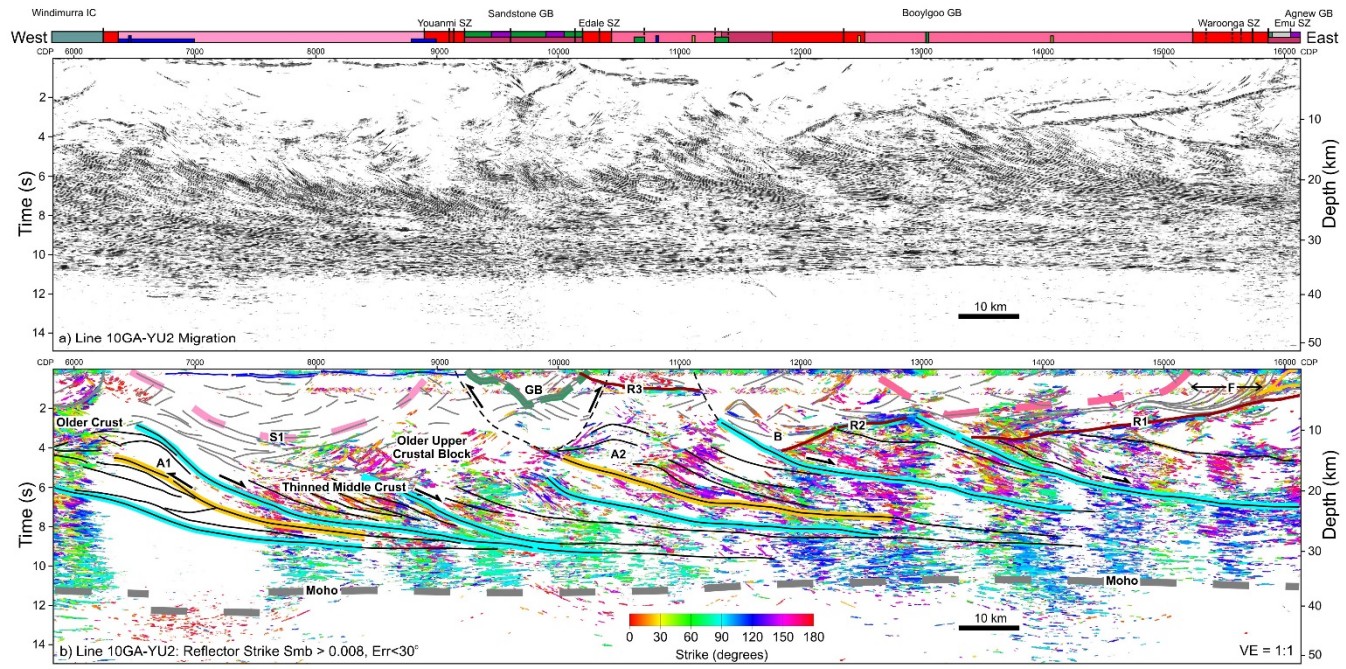

**Figure 3: Line 10GA-YU2: a) F-K migration, b) Migrated reflector strike with interpretation from Calvert and Doublier (2018); only arrivals with semblance greater than 0.008 and strike estimation error less than 30 degrees are included. Pink to dark red – granitoid rocks and gneiss, green, dark green – mafic volcanic rocks, purple – ultramafic volcanic rocks, grey – sedimentary rocks, yellow and blue – Proterozoic sill.**

15   The correspondence between reflections after frequency-wavenumber (F-K) migration and the migrated strike attribute can be assessed by superimposing the strike values on the migrated seismic data, as demonstrated by a section at the east end of line 10GA-YU2 that shows the upper crust near the boundary between the Youanmi Terrane and the Eastern Goldfields Superterrane (Fig. 4). In general, local reflector strike estimates that appear laterally consistent over >2 km overlie clear reflections, but there are many events for which no strike estimates are available due to the limited range of source-receiver

20   azimuths here. Changes in strike value, i.e. colour, along a reflection may indicate variation in the estimated strike due to

limited constraint from the available source-receiver positions within adjacent CDPs or, alternatively, the actual variation in strike due to the geometry of the reflector.

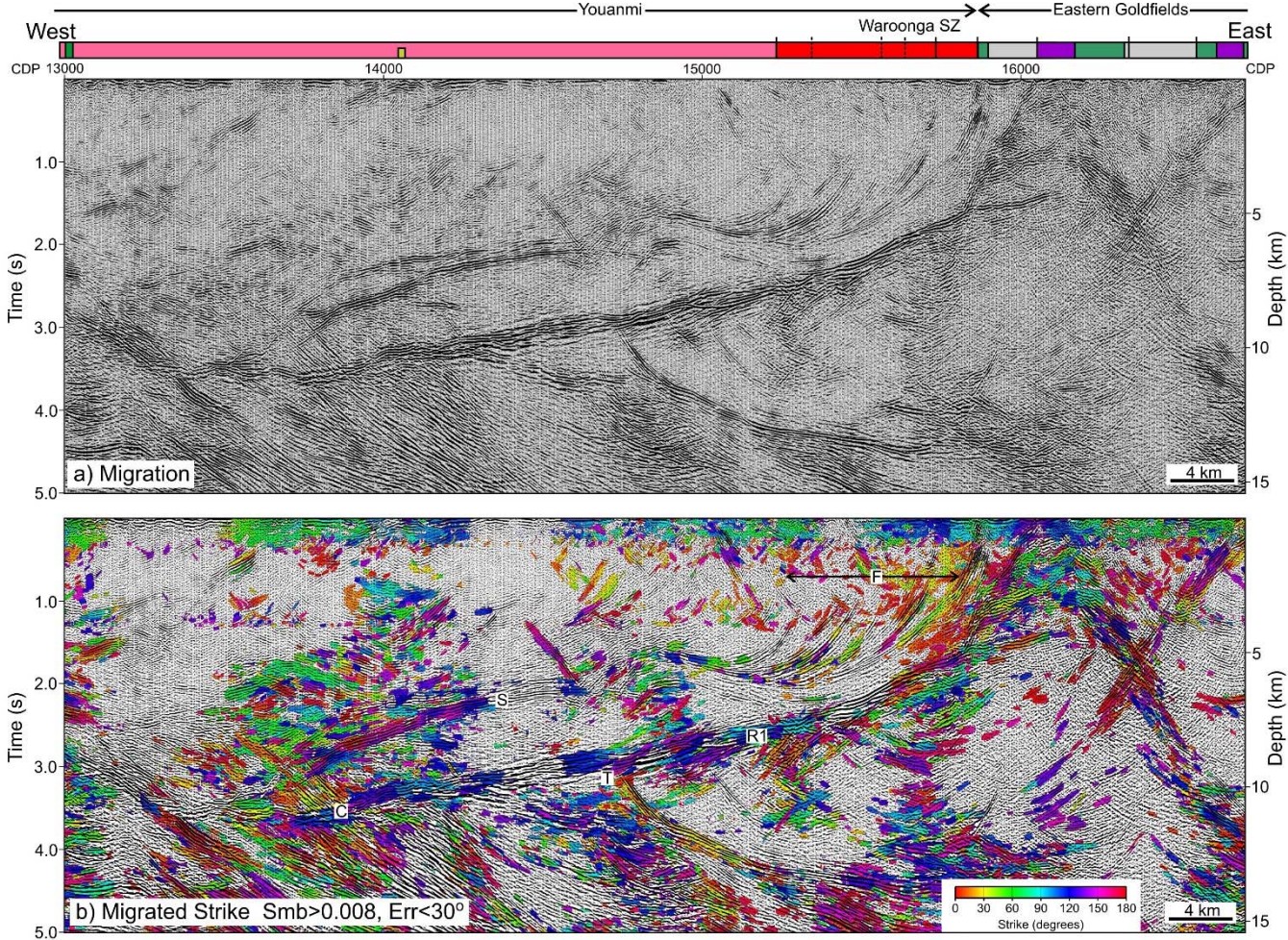

Figure 4: Section of line 10GA-YU2 across the boundary between the Youanmi Terrane and the Eastern Goldfields Superterrane: a) F-K migration, b) Migrated reflector strike superimposed on F-K migration; only arrivals with semblance greater than 0.008 and strike estimation error less than 30 degrees are included. Dark pink – granitoid rocks, red – tonalitic gneiss, green – mafic volcanic rocks, purple – ultramafic volcanic rocks, grey – sedimentary rocks, yellow – Proterozoic sill.

### 3.3 Reflector Strike and Crustal Structure

The estimates of reflector orientation are derived under the assumption that the reflector is locally planar. In the case of complex 3-D structures, for example a dome adjacent to the seismic line, the estimated strike will vary laterally, and inferring the nature of the subsurface structure will not be straight forward. In this situation, one approach would be to model in 3-D the seismic responses of a range of realistic features, estimate the local strikes from the synthetic data, and compare the synthetic results

with the observations. However, where a large region of crust is dominated by reflectors with similar local strikes, then this characteristic, and laterally extensive, reflective fabric is likely to have arisen during a large-scale tectonic process. For example, a tectonic process such as crustal exhumation during shortening or the collapse of thickened crust can produce a thick band of pervasive seismic reflectivity that internally exhibits broadly similar reflector orientations. In this paper, we focus on the identification and interpretation of crustal domains in which a single reflector strike predominates.

Despite the absence of reliable estimates of reflector orientations at many locations along line 10GA-YU2 due to the relatively straight road along which the survey was acquired, it is possible to make some general inferences on the distribution of reflector strike. Specifically, the shallowly dipping reflections in the lower crust between 9 s and 11 s are commonly characterized by values that differ significantly from the overlying middle crust; from CDP 6000-11000 strikes range mostly from 80-110º, and from CDP 11700-16000 values are approximately 90-120º whereas there is a much wider range of values in the middle crust (Fig. 2). Due to the steeper dips of reflections from the middle crust, some of which arrive at times corresponding to the lower crust, it is necessary to interpret them from the migrated section, on which an interpretation (Calvert and Doublier, 2018) has been superimposed (Fig. 3b). In this interpretation anticlines A1 and A2 are inferred to have been formed in an episode of earlier crustal shortening that was then largely overprinted elsewhere in the line by subsequent extension and lower crustal flow. Unfortunately A1 is located in an area of the seismic line where reliable strike estimates could not be recovered, but reflections just to the east associated with, and above, the interpreted thrust have strike values of 0-30º, suggesting that this package of approximately east-dipping mid-crustal reflections between CDP 7300 and CDP 8700 may represent an imbricate stack created in this relatively early thrusting event. Reflections overlying the thrust below A2 exhibit a strike of 120-150º, and may have been created in the same event; however, since this orientation is characteristic of other mid-crustal reflections to the east, it is also possible these reflections were created during the later pervasive collapse of the crust. The listric geometry of many mid-crustal reflections that flatten into the lower crust at different levels between 8s and 10 s has been interpreted as indicating extension and ductile flow of the crust (Calvert and Doublier, 2018); a normal sense of motion is inferred, for example, from the offset of reflections R1 and R2, which appear to occur at the top of the reflective middle crust. Since much of the lower crust exhibits reflector strikes of 80-120º flow is inferred here to be in a N-NNE direction, almost orthogonal to earlier direction of shortening, under the assumption that reflector strike is perpendicular to the direction of ductile flow. The origin of the reflectors is not known, but they could be due to shear zones or syn-tectonic magmatic intrusions whose orientation is controlled by strain fabrics and the stress regime prevailing at that time.

The origin of the large amplitude reflection R1, which is important to the interpretation summarized above, is unclear, because it truncates some underlying reflections, but also appears to cut across others (Fig. 4). After migration, estimates of reflector strike indicate that R1 exhibits a strike of approximately 120º over a distance of more than 20 km; the strike of R1 changes to 000º where it merges with the base of the "fan" of reflections (F in Figure 4b) that project up into the Waroonga shear zone. Both the underlying, abruptly truncated reflections and the cross-cut reflections have a strike of 000-030º (T and C in Figure

4b respectively). Since most reflections above R1 have an opposite sense of apparent dip to those below, R1 represents an angular unconformity, but the high amplitude of R1 and its lateral continuity also suggest that it is a sill. These two perspectives can be reconciled if a sill exploited an existing boundary, perhaps a thrust fault or the base of the brittle upper crust, during its emplacement, but further intruded the package of reflections near its western end, producing the cross-cutting relationship.

There remains, however, some uncertainty due to the fact that while reflections from the approximately north-striking reflectors occur close to the seismic line, those from 120º-striking section of R1 likely occur further away, and are not coincident. Overlying reflection S, which has a fairly consistent strike of 120-130º, may be another subparallel sill that was intruded in the same event. Consequently some reflections may be part of a network of intrusions, perhaps including reflection T, that could have directed melt upward, as has been found by drilling in younger tectonic settings (Juhlin et al., 2016).

**4 Field Recording for Reflector Orientation Estimation**

Along much of line 10GA-YU2, the range of source-receiver azimuths available for the orientation analysis is quite limited, resulting in the exclusion of many estimates due to their large errors. This problem is due to the mostly linear geometry of the road along which the seismic line was shot. Deep seismic lines are typically acquired along existing roads to minimise the cost, which in the case of vibroseis surveys is often determined by the number of shot points that can be acquired per day, i.e.

the source effort. When sufficient recording channels are taken to the field, the incremental cost of deploying additional receivers can be relatively small. If additional recording channels can be placed along crossing roads or readily accessible land through which the survey passes, then the range of available source receiver azimuths can be greatly increased, from <15º to >120º in the synthetic example presented by Calvert (2017). Thus instead of sporadic estimates of strike along a reflection, as shown in Figure 4b, the continuous variation in a reflector's orientation can be determined. This is particularly important when

trying to correlate dipping upper crustal reflections with structures mapped at the surface or trying to distinguish between late sills and the pervasive crustal reflectivity. As an example, the Waroonga shear zone contains steep, north-trending foliations and slivers of greenstone that are subparallel to the pattern of gneissic foliation (Zibra et al., 2017). The shear zone is underlain by a set of reflections that approach the surface with an apparent westerly dip (F in Figure 4b). Perhaps the reflections arise from the contrast between the mafic greenstones and the surrounding more felsic tonalite, but it has not been possible to

confirm that the reflector orientations are consistent with the mapped geological structures due to the limited range of source-receiver azimuths in the seismic survey. In the crystalline basement where reflection geometries can be complex, the availability of complementary reflector orientation attributes can assist an interpretation, perhaps at a basic level by allowing cross-cutting, out-of-plane reflections to be excluded, or potentially by revealing the origin of some enigmatic reflectors in the upper crust.

## 5 Conclusions

In this paper, a method of 2-D line migration that can be applied to any attribute continuously derived from seismic data has been presented. This algorithm uses the apparent dip obtained from the unmigrated stack section to move the attribute to a migrated position where it is represented by short linear segment. (An alternative approach that iterates over an output 3-D migrated volume to identify the most coherent reflections would be much more costly and create artefacts, because an input location can contribute to multiple output locations). Nevertheless the use of reflector orientation information to correctly positon reflectors and their attributes throughout a 3-D volume, perhaps as planar facets, remains a long-term goal, but such an approach requires more accurate orientation estimates, which can be achieved by the use of additional off-line recording during 2-D onshore surveys.

By estimating and migrating the strike of subsurface reflectors along line 10GA-YU2, it has been possible to demonstrate that the lower crust of the eastern Younami Terrane of the Yilgarn Craton exhibits a systematic orientation of shallowly dipping reflectors which mostly dip to the N-NNE or S-SSW, in contrast to the middle crust which is characterized by a broad range of azimuths. Given that much of the crust here has been previously interpreted as reworked during extension and crustal collapse in the Late Archean, we suggest that the orientation of lower crustal reflections is consistent with approximately orogen-normal lower crustal flow at this time.

## Acknowledgements

Tanya Fomin and Ross Costelloe prepared the SEGY files, which included the geometry, binning information necessary for this study. Ross Costelloe and Leoni Jones computed the static corrections during the initial processing that were applied prior to the orientation analysis. We thank Don White and Chris Juhlin for constructive reviews that improved the final manuscript. The field acquisition was funded by Geoscience Australia and the Geological Survey of Western Australia. This reprocessing project was supported by the Natural Sciences and Engineering Council of Canada. M.P.D. is publishing with the permission of the CEO of Geoscience Australia.

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
