# Peer review of "Migration of Reflector Orientation Attributes in Deep Seismic Profiles: Evidence for Decoupling of the Yilgarn Craton Lower Crust"

_Solid Earth, 2019_

## Referee Comment (RC1) · Don White (Referee) · 22 Mar 2019

This paper addresses the interpretation of deep 2D crustal seismic reflection data acquired along crooked acquisition lines. It provides an excellent demonstration of how the crooked nature of the seismic line, which most often is considered a detriment, can be exploited to provide geometrical information that further constrains geological interpretation of the seismic profiles. Specifically, the true orientation (strike and dip) of individual reflectors is determined through stacking analysis for a range of assumed trial strike and dip values. The attribute associated with each coherent reflector (strike

or dip) is then geometrically migrated and overlain on the 2D migrated seismic section for direct comparison. The added value of this methodology is in providing constraints on 3D orientation and subsurface positioning of reflectors that helps clarify geological relationships (cross-cutting fabrics, disconformities etc.)

There are some aspects of the paper that require further justification or clarification. The ones which I would suggest are the following.

Strike and dip as used by the authors refer to local (at the scale of the reflectors) measurements; i.e., the values provide the true local orientation of the reflector. As such, for more complex 3D geometries, it is important to recognize that these are local measurements. In making a structural interpretation, they contribute to an understanding of the true geometry, but don't directly identify unique structural domains. I don't think the authors make this entirely clear, giving too much credence to the resultant strike values (on p2 l4 it is stated that 'Reflector strike is probably the orientation attribute that contributes most to an interpretation').

The authors associate differences in reflector strike directly to different strain regimes (e.g., flow directions, vergence, and age of deformation) which I don't think is an immediately obvious connection. Justification of this underlying assumption should be given or further explanation of the basis for interpretation (observed geometries and relationships between reflection bands or zones) should be stated.

In terms of the algorithm, is there a trade-off between reflector strike and dip determined by the grid search algorithm; i.e., are there multiple values of strike and dip that give the same goodness of fit and how large are the related uncertainties? How does the trade-off depend on the angle of dip? Presumably, the trade-off is quite severe for low dip values (e.g., in the limiting case of a horizontal reflector, the strike is undefined). Can you quantify this with some simple plots (perhaps something like the plots in Levin 1971). Some assessment, or at least a description of some of these issues and the related uncertainties should be included.

As described in the text, the attributes (specifically, reflector strike values) are migrated in 2D as line segments so that they can be overlain on the 2D migrated seismic sections. This is useful. However, it doesn't resolve the old issue of applying 2D migration to crooked line seismic data (which is really 3D). The authors acknowledge this, and note that the facets could actually be migrated in 3D. In contrast to the authors, I think that 3D migration can be quite informative. Perhaps they could make some statement about conditions under which this might be useful.

In regard to attribute migration, the description of the process needs some further elaboration. Currently, the text reads 'Using these apparent dips and the 1-D stacking velocity function, each attribute sample was migrated to a 320 m-long linear segment centred on its output location with only the most coherent event retained at each position, as described earlier.' Why 320 m?

The text states that reflectors with migrated dips of > 50 degrees were excluded in the process. This in conjunction with the postulated large uncertainties (see earlier comment) for very low dips, suggests that the method is most useful for intermediate dips. Is this a fair statement? If so, perhaps should incude this in the conclusions.

At various places in the text, dips are referred to as 'steeply' dipping when they are actually moderately dipping. This should be checked.

For the reader who is interested in applying this method, it would be very useful to provide some guidance on how large a S-R azimuth range is required to get reasonable estimates.

It is stated that 'however, in practice, those parts of the seismic line where it is difficult to obtain orientation angles are reasonably well predicted by the range of available source-receiver azimuths, which is defined to be the number of one degree azimuth bins for which there are seismic data available.' But, isn't it really the azimuth range that is important? For example, cases where there aren't a large number bins, but the actual range is large would give good results?

In the interpretation, only results where uncertainties are < 30 degrees are included. But based on the stated uncertainty range (30 degrees) the lower crust has the same strike values within the uncertainty limits. This would suggest that the method is not particularly useful for the lower crust, but it's real value is for the middle and upper crust. It would be useful to include a plot that shows the uncertainties in section format.

---

## Referee Comment (RC2) · Christopher Juhlin (Referee) · 24 Mar 2019

The authors present an interesting concept on how additional information on reflector orientation may be obtained by taking advantage of crooked line acquisition geometry. The fact that the methodology is automated makes the methodology highly relevant for being employed on an "industrial" scale. This is in contrast to a recent paper that was just accepted in Solid Earth in which the crossdip estimation is manual. The authors may want to reference this paper:

[Figure]

se-2018-120: The crossdip correction as a tool to improve imaging of crooked line seismic data: A case study from the post-glacial Burträsk fault, Sweden, Ruth A. Beckel and Christopher Juhlin

It would be interesting to hear the views of the authors concerning other advantages of their method compared to the "classic" cross-dip correction.

Some other comments:

1. It would be useful to see a geometry plot of the crooked line with examples where the source-receiver azimuths are sufficient for accurate estimation of reflector strike and where they are insufficient.

2. The interpretation presented seems reasonable. However, I wonder if there are more reflections in Figure 4 that can be interpreted as originating from mafic sheets in a granitic host rock. For example, the opposite dipping reflection intersecting the R1 reflection at point "T" could represent what was once a near-vertical feeder dike that has been deformed. The entire area may have been intruded by a generation of horizontal and vertical mafic sheets that were later deformed. This would be a similar situation as we may have in the more modern Scandinavian Caledonides. See

Juhlin et al., 2016. Seismic imaging in the eastern Scandinavian Caledonides: siting the 2.5 km deep COSC-2 borehole, central Sweden. Solid Earth, 7, 769–787.

3. The authors mention 3D pre-stack migration as an option, but discount it based on the assumption that the data are too sparse. Another option would be to perform 3D binning and process to stack to map the orientation of some reflections. Was this tried? The paper below shows a successful example in locating an off-profile diffractor:

Malehmir, A., Schmelzbach, C., Bongajum, E., Bellefleur, G., Juhlin, C., and Tryggvason, A., 2009. 3D constraints on a possible deep >2.5 km massive sulphide mineralization from 2D crooked-line seismic reflection data in the Kristineberg mining area, northern Sweden. Tectonophysics, 479, 223–240.

---

## Author Comment (AC1) · 18 Apr 2019

Comment 1: Dip and strike are local measurements, and relation between strike and different strain regimes is not immediately obvious, requiring further justification.

Response 1: The reviewer makes a good point about the individual estimates of strike being local in nature. We have modified the statement identified as p2 l4, and added an explanatory paragraph at the start of section 3.3 to discuss how similar strike estimates across a larger crustal domain can be indicative of a larger scale tectonic process.

Comment 2: What is the nature of the trade-off between dip and strike, and what are the uncertainties?

Response 2: For an individual CDP there is usually a trade-off between local dip and strike values due to the limited range of available source-receiver azimuths, and this issue has been previously documented by Bellefleur et al. (1997) in their Fig. 4 and Calvert (2017), also in Fig. 4. But as these authors show, this problem can be addressed by including more CDP gathers within the CDP supergather that is used for the analysis, often producing a well-defined global maximum in the estimated semblance; in the case of line YU2, 64 CDP were combined into a single supergather. The methodology section has been expanded to make this clearer. Other than the simple case shown by Levin (1971), it is not possible with a crooked seismic line to produce a simple plot to characterize this trade-off, because the trade-off also depends on the midpoint location, which varies for each trace, in contrast to a straight line. We have, however, taken the approach of estimating the range of angles within 90% of the global semblance maximum to provide an estimate of the relative error for every determined strike value. We have added a display of the error estimated in this way to Figure 2.

Comment 3: 3D migration can be quite informative, can the authors suggest when this might be useful?

Response 3: A general statement on the value of 3D migration has been added at the start of section 2.1, but it is difficult to be more specific, because the value of low fold 3-D migration depends on a number of factors, including the acquisition geometry, signal-to-noise ratio etc. We do state that some form of 3D migration is the desirable goal, and now cite a paper by Nedimovic and West (2003b) that investigates this issue for a Lithoprobe crooked line geometry.

Comment 4: Attribute migration requires further elaboration and explanation for choice of 320 m segment length.

Response 4: A statement has been added to explain and justify the choice of the 320

m length for the output migrated segment, explaining the method in a bit more detail.

Comment 5: Is attribute migration most suitable for intermediate dips?

Response 5: Migrated dips were restricted to 50 degrees to limit the inclusion of steeply dipping coherent noise, and this is now explained. Dips up to 80 or so degrees can be readily migrated, but it can also become difficult to incorporate steep reflections events into an interpretation when they are out of plane, because they migrate over quite large distances. It should be emphasised that, as noted above, low dip values can usually be estimated accurately when sufficient source-receiver azimuths are present, e.g. with a large enough supergather, along a crooked line. Of course this is not possible when the line is perfectly straight. At very small dips, there may also be a larger error in the strike, but these events can still be migrated, and will not move far due to their low apparent dip.

Comment 6: Change steeply dipping to moderately dipping where appropriate.

Response 6: The text has been modified accordingly.

Comment 7: What is the range of source-receiver azimuths typically required for reasonable orientation estimates, and why is range estimated using a binned measure?

Response 7: The minimum range of source-receiver azimuths required for accurate orientation estimates, 20-30 degrees, is now included. We have also explained in the text that we use a binned estimate of source receiver azimuth range to avoid cases where a large azimuth range is created by a single orthogonal source-receiver pair that will not contribute much to the resolving reflector orientation due to the low signal to noise of a single trace.

Comment 8: Do the uncertainties make the method less useful in the lower crust, and can a plot of the uncertainties in section format be included?

Response 8: We have included an additional display in Figure 2 to show the errors in the estimated strike values for reflections with a semblance greater than 0.005. Al-
though the display in Figure 2a, includes reflections with errors in strike of less than 30 degrees, it can be seen that many reflections in the lower crust have much lower errors, commonly less than 10 degrees where included in Figure 2a. The method we outline is not limited to particular depth levels of the crust. The effectiveness of the method appears to be controlled mostly by the linearity of the seismic acquisition geometry.

References

Bellefleur, G., Calvert, A.J., and Chouteau, M.C.: A link between deformation history and the orientation of reflective structures in the 2.68-2.83 Ga Opatica belt of the Superior Province, J. Geophys. Res., 102, 15243-15257, doi:10.1029/97jB00505, 1997.

Calvert, A.J.: Continuous estimation of 3-D reflector orientations along 2-D deep seismic reflection profiles, Tectonophysics, 718, 61-71, doi:10.1016/j.tecto.2016.11.002, 2017. Levin, F.K.: Apparent velocity from dipping interface reflections, Geophysics, 36, 510-516, 1971.

Nedimović, M.R. and West, G.F.: Crooked-line 2D seismic reflection imaging in crystalline terrains: Part 2, migration, Geophysics, 68, 286-296, 2003b.

---

## Author Comment (AC2) · 18 Apr 2019

Comment 0: What are the advantages of the automated method compared with the traditional manual approach?

Response 0: We now include a brief comparison of the automated reflector orientation estimation method with manual cross-dip selection, and reference the paper by Beckel and Juhlin (2018).

Comment 1: It would be useful to see a plot of crooked line with examples of where

the source-receiver azimuths are sufficient for the estimation.

Response 1: We have not included a map of the line in the paper, because it does not really add that much to the basic map in Figure 1; however, for the information of the reviewers we include in our response here a more detailed display of the acquisition profile corresponding to the seismic data shown in Figure 2. We have also now included a section in Figure 2b showing the strike estimation error, indicating where source-receiver azimuths are insufficient for good orientation estimates.

Comment 2: Can more reflections be interpreted as intrusions with T being an old feeder dyke?

Response 2: We now note that reflection T could be a feeder dyke, and mention the possibility that some of the imaged reflections arise from later intrusions. We also cite the paper by Juhlin et al. (2016).

Comment 3: Was 3D binning followed by stack tried to determine the orientation of some reflections?

Response 3: We did not stack the data after 3D binning to estimate reflector orientations. One important limitation of this approach is that the stacking velocity required for an accurate moveout correction is azimuth and dip dependent. 3-D DMO would help with this problem. However, by estimating the reflector orientations directly from the prestack data, as we have done, limitations due to having to apply a moveout correction are avoided.

References

Beckel, R.A., and Juhlin, C.:The crossdip correction as a tool to improve imaging of crooked line seismic data: A case study from the post-glacial Burtrask fault, Sweden, Solid Earth, doi:10.5194/se-2018-120, 2018.

Juhlin, C., Hedin, P., Gee, D.G., Lorenz, H., Kalscheuer, T., and Yan, P.: Seismic imaging in the eastern Scandinavian Caledonides: siting the 2.5 km deep COSC-2

borehole, central Sweden, Solid Earth, 7, 769-787, 2016.

[Figure]

**Fig. 1.**